# An Intrinsic Characterization of Shannon’s and Rényi’s Entropy

**DOI:** 10.3390/e26121051

**Published:** 2024-12-04

**Authors:** Martin Schlather, Carmen Ditscheid

**Affiliations:** Institute of Mathematics, University of Mannheim, 68131 Mannheim, Germany

**Keywords:** chain rule, characterization, Hartley entropy, min-entropy, Rényi entropy, Shannon entropy

## Abstract

All characterizations of the Shannon entropy include the so-called chain rule, a formula on a hierarchically structured probability distribution, which is based on at least two elementary distributions. We show that the chain rule can be split into two natural components, the well-known additivity of the entropy in case of cross-products and a variant of the chain rule that involves only a single elementary distribution. The latter is given as a proportionality relation and, hence, allows a vague interpretation as self-similarity, hence intrinsic property of the Shannon entropy. Analogous characterizations are given for the Rényi entropy and its limits, the min-entropy and the Hartley entropy.

## 1. Introduction

The Shannon entropy *H* has been fully characterized by different sets of conditions [1,2,3,4]. The conditions themselves have been investigated deeply by [5,6], for instance. All characterizations use variants of the so-called chain rule, which can be summarized as follows. Let $p=\{p_{1},\ldots ,p_{n}\}$ and $q=\{q_{1},\ldots ,q_{m}\}$ be discrete probability distributions, and
(1)$$\begin{array}{c} p_{k,q}=\left\{p_{1},\ldots ,p_{k-1},p_{k}q_{1},\ldots ,p_{k}q_{m},p_{k+1},\ldots ,p_{n}\right\}. \end{array}$$The chain rule states that
(2)$$\begin{array}{c} H\left(p_{k,q}\right)=H\left(p\right)+p_{k}H\left(q\right). \end{array}$$The construction in Equation (Equation 1) can be iterated, i.e., *p* is extended at several positions $k_{i}$ with different distributions $q^{\left(i\right)}$. In case of two positions, this is
$$p_{k_{1},q^{\left(1\right)};k_{2},q^{\left(2\right)}}:={\left(p_{k_{2},q^{\left(2\right)}}\right)}_{k_{1},q^{\left(1\right)}},$$
where $k_{1}<k_{2}$. The most elementary version of the chain rule is due to Faddeev [2], where *q* in Equation (Equation 1) is restricted to a Bernoulli distribution. Carcassi et al. [7] considered the full version of Equation (Equation 1), where *q* is arbitrary, as preferable from a didactical point of view, since this allows an immediate interpretation in various areas of application. Often, the completely iterated version,
(3)$$\begin{array}{c} H\left(p_{1,q^{\left(1\right)};\ldots ;n,q^{\left(n\right)}}\right)=H\left(p\right)+\sum_{k=1}^{n}p_{k}H\left(q^{\left(k\right)}\right), \end{array}$$
is taken [4], since it allows the practical interpretation that first an alphabet $\alpha_{k}$ is chosen with probability $p_{k}$ and subsequently a letter within $\alpha_{k}$ according to $q^{\left(k\right)}$.

Baez et al. [8] give the first algebraic approach, namely, in terms of information loss. The latter is defined as $F_{p,q}=H\left(p\right)-H\left(q\right)$, where the probability distribution *q* is a function of *p*, hence $F_{p,q}$ quantifies the change in entropy of the transformation. The information loss $F_{p,q}$ is uniquely characterized by a number of properties, including $F_{p,q}$, being a convex linear map, i.e., H(λp∪(1−λ)q)=λH(p)+(1−λ)H(q), which might be considered as a restatement of Faddeev’s chain rule.

Common to all versions of the chain rule is that they describe situations where, canonically, at least two different distributions, *p* and *q*, are involved. Ebanks et al. ([6], Definition 3.1.1) introduced an alternative notation of Faddeev’s condition, which formally considers only one distribution but, nevertheless, relies on computing the Shannon entropy of two distributions, namely, *p* and a Bernoulli distribution, to state the chain rule.

The Rényi entropy generalizes the Shannon entropy and can be fully characterized via quasi-arithmetic means ([5], Section 5). Another approach to characterize the Rényi entropy relies on a transform. The order-α information energy *S* with Onicescu’s information energy as a special case [9] is, up to a multiplicative constant, the sum of the α-powers of the probabilities. Pardo [10] gives a full characterization by means of a chain rule that is similar to Faddeev’s variant,
(4)$$\begin{array}{c} S\left(p_{k,q}\right)=S\left(p\right)+p_{k}^{\alpha }\left(S\left(q\right)-\mathrm{const}\right), \end{array}$$
where *q* is a Bernoulli distribution and the constant depends on α and the normalization of *S*. Pardo [10] also generalizes this equation by introducing weighted sums. Leinster ([4], Theorem 4.5.1) characterizes the information energy by a formula that is analogous to the fully iterated version of the chain rule (Equation 3).

The limits of the Rényi entropy as its parameter goes to 0 and *∞* are the Hartley entropy [11] and the min-entropy [12], respectively. Mathematical characterizations of these two entropies are rare, e.g., conical combinations of the Hartley entropy and the Shannon entropy are characterized in [13] as the only reasonable simultaneous solution to the chain rule and the subadditivity property. Ref. [13] also characterized the Hartley entropy as being insensitive. An advanced investigation of the min-entropy exists in form of the Fourier min-entropy [14].

Schlather [15] deals with a rather general approach to the entropy, which is axiomatically based on the additivity assumption of the entropy for independent systems. Under certain conditions, called scale invariance there, the entropy is unique. This result, however, is not applicable here, since it is closely related to scalar real numbers and not to probability distributions. Yet, the characterizations given here are in this spirit.

To be specific, let q=p in Equation (Equation 1). Then, we obtain
(5)$$\begin{array}{c} p_{k,p}=\left\{p_{1},\ldots ,p_{k-1},p_{k}p_{1},\ldots ,p_{k}p_{n},p_{k+1},\ldots ,p_{n}\right\}, \end{array}$$
and, hence, a proportionality relation for the Shannon entropy by Equation (Equation 2),
$$\begin{array}{c} H\left(p_{k,p}\right)=\left(1+p_{k}\right)H\left(p\right). \end{array}$$Theorem 1 in Section 2 states that the chain rule in Khinchin’s (1953) characterization [1,2] can be split into two canonical components, the additivity property and the proportionality between $H(p_{k,p})$ and H(p). The characterization of the Rényi entropy is in analogy to the characterization of the Shannon entropy, except that the functional Equation (Equation 4), i.e.,
$$\begin{array}{c} S\left(p_{k,p}\right)=\left(1+p_{k}^{\alpha }\right)S\left(p\right)-p_{k}^{\alpha }, \end{array}$$
reflects the proportionality relation. Related characterizations also exist for the min-entropy and the Hartley entropy, cf. Theorems 3 and 4. All technical parts are postponed to Section 3. There, the geometric distribution plays an overwhelming role. Section 4 illuminates practical and theoretical aspects of this paper. Section 5 finalizes with a digest of our approach.

## 2. Results

We denote by {.} an ordered set of non-negative numbers and let
(6)$$\begin{array}{ccc} a\{p_{1},p_{2},\ldots \} & = & \{ap_{1},ap_{2},\ldots \},\;a\geq 0,\\ \left\{p_{1},p_{2},\ldots \right\}\cup \left\{q_{1},q_{2},\ldots \right\} & = & \{p_{1},p_{2},\ldots ,q_{1},q_{2},\ldots \}. \end{array}$$As we will address only the very first elements of an ordered set, the intuitive, but sloppy, notation on the right-hand side of Equation (Equation 6) is sufficient.

Let us denote by *P* the set of all discrete probability distributions and by $P_{n}\subset P$ the set of distributions with at most *n* atomic events that have a probability greater than zero. We denote such a probability distribution by $\left\{p_{1},\ldots ,p_{n}\right\}\in P_{n}$. The uniform distribution is denoted by $U_{n}=\left\{1/n,\ldots ,1/n\right\}$. Let $P_{0}={⋃}_{n}P_{n}$ be the set of all discrete probability distributions with finite support.

Reordering the elements in Equation (Equation 5) will turn out to be useful in the proofs. We define
(7)$$\begin{array}{ccc} p_{\hat{k}} & = & \{p_{i}:i\neq k\},\\ \partial_{k}p & = & p_{k}p\cup p_{\hat{k}}, \end{array}$$
where $p=\{p_{1},p_{2},\ldots \}\in P$. For instance,
$$\begin{array}{ccc} \partial_{3}\left\{p_{1},p_{2},p_{3}\right\} & = & \{p_{3}p_{1},p_{3}p_{2},p_{3}p_{3},p_{1},p_{2}\}. \end{array}$$We use the convention 0log0=0.

**Theorem** **1.**
*Let H:P→[0,∞] be a function. Then, the following two assertions are equivalent:*
*1.* 
*H is the Shannon entropy,*

$$H\left(p\right)=-\sum_{p_{i}\in p}p_{i}\log p_{i},$$

*up to a positive multiplicative constant.*
*2.* 
*H has the following properties:*
*(a)* 
*H is a continuous function in the topology of convergence in distribution;*
*(b)* 
*$H\left(\left\{p_{1},\ldots ,p_{n}\right\}\right)=H\left(\left\{p_{\pi (1)},\ldots ,p_{\pi (n)}\right\}\right)$ for $\left\{p_{1},\ldots ,p_{n}\right\}\in P_{0}$ and any permutation π;*
*(c)* 
*$H\left(\left\{p_{1},\ldots ,p_{n}\right\}\right)=H\left(\left\{0,p_{1},\ldots ,p_{n}\right\}\right)$ for all $\left\{p_{1},\ldots ,p_{n}\right\}\in P_{0}$;*
*(d)* 
*$H\left(p\right)\leq H\left(U_{n}\right)<\infty$ for all $p\in P_{n}$, n∈N;*
*(e)* 
*H(p)∈(0,∞) if p is a nondegenerate geometric distribution;*
*(f)* 
*H(p×q)=H(p)+H(q) for all p,q∈P;*
*(g)* 
*a function f:[0,1]→[0,∞) exists such that*

(8)
$$\begin{array}{c} H\left(\partial_{1}p\right)=\left(1+f\left(p_{1}\right)\right)H\left(p\right),\;p\in P. \end{array}$$




Conditions 2(a)–2(g) can be regarded as interpretable, hence intrinsic properties ([5], Section 1.2). For this reason, we prefer Condition 2(d) over a monotonicity assumption on *H*. Note that Conditions 2(a)–2(d) are Khinchin’s (1953) conditions as given in [2] except for the chain rule, which is replaced by Conditions 2(e)–2(g).

**Remark** **1.***Note that, here and in the subsequent theorems, the function f is not unique, since it is undetermined at* 1. *A convenient choice is to take $f\left(1\right)=\lim_{a\to 1}f\left(a\right)$.*

**Remark** **2.**
*Our approach has a simple, but nice implication. Let $p,q\in P_{0}$. Then,*

$$\begin{array}{ccc} \mathrm{KL}(\partial_{1}p,\partial_{1}q) & = & \left(1+p_{1}\right)\mathrm{KL}\left(p,q\right), \end{array}$$

*where KL denotes the Kullback–Leibler divergence.*


**Theorem** **2.**
*Let H:P→[0,∞] be a function. Then, the following two assertions are equivalent:*
*1.* 
*H is the Rényi entropy,*

$$H\left(p\right)=\frac{1}{1-\alpha }\log\sum_{p_{i}\in p}p_{i}^{\alpha },$$

*for some α∈(0,∞)∖{1}, up to a positive multiplicative constant.*
*2.* *H satisfies Conditions* 2(a)–2(f) *of Theorem 1. A value γ∈R∖{0} and a function f:[0,1]→[0,∞) exist such that for $S=e^{\gamma H}$ we have*(9)$$\begin{array}{c} S\left(\partial_{1}p\right)=\left(1+f\left(p_{1}\right)\right)S\left(p\right)-f\left(p_{1}\right),\;p\in P. \end{array}$$


We denote the max-operator by ∨ and the min-operator by ∧.

**Theorem** **3.**
*Let H:P→[0,∞] be a function. Then, the following two assertions are equivalent:*
*1.* 
*H is the min-entropy,*

$$H\left(p\right)=-\log\underset{p_{i}\in p}{⋁}p_{i},$$

*up to a positive multiplicative constant.*
*2.* *H satisfies Conditions* 2(a)–2(f) *of Theorem 1. A function f:[0,1]→[0,∞) exists, such that for $S=e^{-H}$, we have*(10)$$\begin{array}{c} f\left(p_{1}\right)\vee S\left(\partial_{1}p\right)=S\left(p\right),\;p\in P. \end{array}$$


As the Hartley entropy *H* is discontinuous on *P* and $H(g_{a})=\infty$, Conditions 2(a) and 2(e) are replaced by Equations (Equation 11) and (Equation 12), respectively.

**Theorem** **4.**
*Let H:P→[0,∞] be a function. Then, the following two assertions are equivalent:*
*1.* 
*H is the Hartley entropy,*

$$H\left(p\right)=\log\sum_{p_{i}\in p}1_{\left(0,1\right]}\left(p_{i}\right),$$

*where $1_{\left(0,1\right]}\left(p_{i}\right)=1$ if $p_{i}\in \left(0,1\right]$, and 0 otherwise.*
*2.* *H satisfies Conditions* 2(b)–2(d) *and* 2(f) *of Theorem 1 and $H\left(U_{2}\right)\in \left(0,\infty \right)$. Further, for $S=e^{H}$ holds*(11)$$\begin{array}{c} \lim_{n\to \infty }\frac{S(\left\{p_{1},\ldots ,p_{n}\right\}\cup q\sum_{i=n+1}^{\infty }p_{i})}{S(\left\{p_{1},\ldots ,p_{n}\right\}/\sum_{i=1}^{n}p_{i})}\in \left\{1,\infty \right\},\;\left\{p_{1},p_{2},\ldots \right\},q\in P. \end{array}$$*Moreover, S(p)<∞,p∈P with $p_{1}<1$ implies $S(p_{\hat{1}}/\left(1-p_{1}\right))<\infty$. A function f:[0,1]→[0,∞) exists such that Equation* (Equation 9) *holds true and*(12)$$\begin{array}{c} S\left(\frac{{\left(\partial_{1}p\right)}_{\hat{1}}}{1-p_{1}^{2}}\right)=\left(1+f\left(p_{1}\right)\right)S\left(\frac{p_{\hat{1}}}{1-p_{1}}\right),\;p\in P,p_{1}<1. \end{array}$$


## 3. Proofs

We use the following abbreviations:$$\begin{array}{ccc} F_{n}\left(a\right) & = & \prod_{k=0}^{n-1}\left(1+f(a^{2^{k}})\right),\;a\in \left[0,1\right],\\ F(a) & = & \prod_{k=0}^{\infty }\left(1+f(a^{2^{k}})\right),\;a\in \left(0,1\right). \end{array}$$

### 3.1. Functional Equations

**Lemma** **1.**
*Let f be a real-valued function on (0,1). The following three assertions are equivalent:*
*1.* 
*f is continuous and, for all a∈(0,1), we have*

(13)
$$\begin{array}{ccc} f(a)+f(1-a) & = & 1 \end{array}$$


(14)
$$\begin{array}{ccc} \left(1-f(a))F(a\right) & = & 1; \end{array}$$

*2.* *f is continuous and, for all a∈(0,1), Equation* (Equation 13) *holds and $f\left(a^{2}\right)=f^{2}\left(a\right)$;**3.* 
*f is the identity.*



**Proof.** Let *f* be the identity. Equation 0.266 in [16] shows that Equation (14) holds true. The other properties are obvious.Now, assume that the first condition applies. Then, the second condition holds true, since
$$\begin{array}{ccc} 1 & = & \left(1-f\left(a\right)\right)\left(1+f\left(a\right)\right)\prod_{k=1}^{\infty }\left(1+f\left(a^{2^{k}}\right)\right)\\ & = & \left(1-f\left(a\right)\right)\left(1+f\left(a\right)\right)/\left(1-f\left(a^{2}\right)\right), \end{array}$$
from which the equation $f\left(a^{2}\right)=f^{2}\left(a\right)$, a∈(0,1), follows.We now show that the second condition implies that *f* is the identity. For an endomorphism *h*, we denote by $h^{\left(n\right)}$ the *n*-fold composition h∘…∘h. The assumptions imply that $f\left({\left(1-a^{2^{k}}\right)}^{2^{r}}\right)={\left(1-f^{2^{k}}\left(a\right)\right)}^{2^{r}}$ for all a∈(0,1) and k,r∈Z. Note that the set of maps $\left\{a\mapsto {\left(1-a^{2^{k}}\right)}^{2^{r}}:k,r\in Z\right\}$ is identical to the set of its inverses. Equation (Equation 13) yields f(1/2)=1/2. Let $T\left(A\right)=\{{\left(1-a^{2^{k}}\right)}^{2^{r}}:a\in A,r,k\in Z\}\cup A$ for A⊂(0,1), and $D={⋃}_{n}T^{\left(n\right)}\left(\left\{1/2\right\}\right)$. It suffices to show that *D* is dense in (0,1). Assume that it is not, i.e., an interval (a,b)⊂(0,1) exists with D∩(a,b)=∅. Then, necessarily, also $D\cap {⋃}_{n}T^{\left(n\right)}\left(\left(a,b\right)\right)=\emptyset$. Since 1/2∈D, we may assume, without loss of generality, that (a,b)⊂(1/2,1). Consider the map *h* defined on the open intervals of (0,1),
$$h\left(\left(a,b\right)\right)\mapsto \left\{\begin{array}{cc} (a^{2},b^{2}), & \mathrm{if}a\geq 1/2,b^{2}>1/2\\ (1-b^{2},1-a^{2}), & \mathrm{if}a\geq 1/2,b^{2}\leq 1/2\\ (a,b), & \mathrm{if}a<1/2 \end{array}\right..$$Note that $\left(1-b^{2},1-a^{2}\right)\subset T\left(\left(a,b\right)\right)$ and $\left(a^{2},b^{2}\right)\subset T^{\left(2\right)}\left(\left(a,b\right)\right)$. Let $\left(c,d\right)=h^{\left(n-1\right)}\left(\left(a,b\right)\right)$. Per assumption, *h* maps (a,b)⊂(1/2,1) to either $\left(a^{2},b^{2}\right)$ or $\left(1-b^{2},1-a^{2}\right)$. Per construction, if $h^{\left(n\right)}\left(\left(a,b\right)\right)=\left(1-c^{2},1-d^{2}\right)$, then $h^{\left(n+1\right)}\left(\left(a,b\right)\right)\neq \left(1-c^{2},1-d^{2}\right)$. If, eventually, $h^{\left(n\right)}\left(\left(a,b\right)\right)=h^{\left(n+1\right)}\left(\left(a,b\right)\right)=\left(c^{2},d^{2}\right)$, then it always holds that $1/2\in (c^{2},d^{2})$. Let |(a,b)| be the length of an interval (a,b). If a≥1/2, then
$$\left|h\left(a,b\right)\right|=b^{2}-a^{2}=\left(b+a\right)\left(b-a\right)>b-a.$$Hence, $h^{\left(n\right)}\left(\left(a,b\right)\right)$ finally contains 1/2 or its length is strictly monotonously increasing. If the latter were true, the limit would exist as |h|≤1/2. Close to the limit, we would have $|h^{\left(n+1\right)}\left(\left(a,b\right)\right)-h^{\left(n\right)}\left(\left(a,b\right)\right)|=|d^{2}-c^{2}-\left(d-c\right)|=|\left(d-c\right)\left(d+c-1\right)|<\epsilon$ for some ϵ>0, i.e., both boundaries of $h^{\left(n\right)}\left(\left(a,b\right)\right)$ must converge to 1/2, contradicting the fact that an interval with upper bound close to 1/2 is mapped to an interval with upper bound close to 3/4. □

**Lemma** **2.**
*Let Φ:[0,1]→[0,1] be an endomorphism and c>0. The following two assertions are equivalent:*
*1.* Φ *is continuous on (0,1) with values in (0,1), Φ(1/2)>0 and Φ(1/n)→0=Φ(0) as n→∞. For all n∈N and 0≤k<n, we have*
$$\begin{array}{ccc} \Phi \left(a^{n}\right)\vee \Phi \left(1-a\right) & = & \Phi^{n}\left(a\right)\vee \Phi \left(1-a\right),\;a\in \left(0,1\right),\\ \frac{\Phi (1-k/n)}{{\left(n-k\right)}^{c}}\vee \Phi \left(k/n\right) & = & \prod_{m=k+1}^{n}\Phi \left(1-1/m\right)\vee \overset{n}{\underset{\ell =k+1}{⋁}}\Phi \left(1/\ell \right)\prod_{m=\ell +1}^{n}\Phi \left(1-1/m\right); \end{array}$$*2.* 
*$\Phi \left(a\right)=a^{c}$.*



**Proof.** We assume that the first condition holds true. To begin with, we show that Φ is strictly increasing. For j∈N, we have
$$\frac{\Phi (1-k/n)}{j^{c}{\left(n-k\right)}^{c}}\vee \Phi \left(k/n\right)=\prod_{m=jk+1}^{jn}\Phi \left(1-1/m\right)\vee \overset{jn}{\underset{\ell =jk+1}{⋁}}\Phi \left(1/\ell \right)\prod_{m=\ell +1}^{jn}\Phi \left(1-1/m\right).$$Letting j→∞, we obtain
$$\Phi \left(k/n\right)=\lim_{j\to \infty }\prod_{m=jk+1}^{jn}\Phi \left(1-1/m\right),$$
so that Φ(k/n)≤Φ((k+1)/n) for all 0<k<n−1. Hence, Φ is increasing on (0,1)∩Q, thus increasing on (0,1). In particular, Φ≥Φ(1/2)>0 on [1/2,1). Let n/2≤k<n−1 and j∈N such that $j^{c}{\left(n-k-1\right)}^{c}\Phi \left(k/n\right)>1$. Then,
$$\begin{array}{ccc} \Phi ((k+1)/n) & = & \Phi \left(1-\left(k+1\right)/n\right)j^{-c}{\left(n-k-1\right)}^{-c}\vee \Phi \left(\left(k+1\right)/n\right)\\ & = & \prod_{m=j(k+1)+1}^{jn}\Phi \left(1-1/m\right)\\ & > & \prod_{m=jk+1}^{jn}\Phi \left(1-1/m\right)=\Phi \left(k/n\right), \end{array}$$
so that Φ is strictly increasing on [1/2,1). Therefore, $\Phi \left(a^{n}\right)=\Phi^{n}\left(a\right)$ for all a∈(0,1) and n∈N with $1-a<a^{n}$. Let m,k∈N. Let *ℓ* be a multiple of *k* such that $\left(a\wedge a^{m/k}\right)+a^{1/\ell }-1>0$. Then,
$$\begin{array}{c} \Phi \left(a^{m/k}\right)=\Phi \left({\left(a^{1/\ell }\right)}^{m\ell /k}\right)={\left(\Phi \left(a^{1/\ell }\right)\right)}^{m\ell /k}={\left({\left(\Phi \left(a^{1/\ell }\right)\right)}^{\ell }\right)}^{m/k}={\left(\Phi \left(a\right)\right)}^{m/k}. \end{array}$$The continuity of Φ implies that $\Phi \left(a^{x}\right)=\Phi {\left(a\right)}^{x}$ for a∈(0,1) and x>0, so that $\Phi \left(a\right)=a^{\gamma }$ for a γ∈R. Since Φ(1/n)→0 and Φ(1/2)>0, we have γ>0. Finally, for k=0, we obtain from the assumptions that $\Phi \left(1\right)/n^{c}=1/n^{\gamma }$, n∈N, so that Φ(1)=1 and c=γ. □

Finally, we restate two results from the literature for the reader’s convenience.

**Lemma** **3.***Let H:P→[0,∞] be a function fulfilling Conditions* 2(d) *and* 2(f) *of Theorem 1. Then, $H(U_{n})=c\log n$ for all n∈N and some c≥0.*

**Proof.** The inclusion $P_{n}\subset P_{n+1}$ and Condition 2(d) immediately imply monotonicity, i.e., $H\left(U_{n}\right)\leq H\left(U_{n+1}\right)$ for all n∈N. By Condition 2(f) and Implication 1 of Theorem V in [17], the claim is followed. □

**Lemma** **4.**
*Let H be the Shannon entropy, the Rényi entropy, the min-entropy, or the Hartley entropy. Then, $H(U_{n})$ is maximal on $P_{n}$.*


See, for instance, Lemma 2.2.4 and Remark 4.4.4 in [4] for proofs.

### 3.2. Properties of the Operator $\partial_{k}$

We will use the operator $\partial_{k}$, defined in Equation (Equation 7), iteratively. For instance,
$$\begin{array}{ccc} \partial_{2}\partial_{k}p & = & \left(p_{2}p_{k}\right)\left(p_{k}p\cup p_{\hat{k}}\right)\cup {\left(p_{k}p\cup p_{\hat{k}}\right)}_{\hat{2}},\\ \partial_{2}\partial_{3}\left\{p_{1},p_{2},p_{3}\right\} & = & \partial_{2}\left\{p_{3}p_{1},p_{3}p_{2},p_{3}p_{3},p_{1},p_{2}\right\}\\ & = & \;\;\;\begin{array}{ccccc} \{p_{3}p_{2}p_{3}p_{1}, & \;\;\;\;p_{3}p_{2}p_{3}p_{2}, & \;\;\;\;p_{3}p_{2}p_{3}p_{3}, & \;\;\;\;p_{3}p_{2}p_{1}, & \;\;\;\;p_{3}p_{2}p_{2},\\ p_{3}p_{1}, & & p_{3}p_{3}, & p_{1}, & p_{2}\}\;\;\; \end{array}\\ & = & \{p_{1}p_{2}p_{3}^{2},p_{2}^{2}p_{3}^{2},p_{2}p_{3}^{3},p_{1}p_{2}p_{3},p_{2}^{2}p_{3},p_{1}p_{3},p_{3}^{2},p_{1},p_{2}\}. \end{array}$$The proofs of the theorems in Section 2 rely on the properties of various probability distributions. We denote by $b_{a}$ the Bernoulli distribution, i.e.,
$$b_{a}=\left\{a,1-a\right\},\;a\in \left(0,1\right),$$
and by $g_{a}$ the geometric distribution, i.e.,
$$g_{a}=\left(1-a\right)\left\{1,a,a^{2},\ldots \right\},\;a\in \left[0,1\right).$$For n∈N, let
$$\begin{array}{ccc} g_{n,a} & = & \frac{1-a}{1-a^{n}}\left\{1,a,a^{2},\ldots ,a^{n-1}\right\},\;a\in \left[0,1\right), \end{array}$$
(15)$$\begin{array}{ccc} G_{n,a} & = & \left\{a^{n}\right\}\cup \left(1-a\right)\left\{1,a,a^{2},\ldots ,a^{n-1}\right\},\;a\in \left[0,1\right], \end{array}$$
(16)$$\begin{array}{ccc} U_{k,n} & = & \{1-k/n,\underset{k\;\mathrm{times}}{\underset{︸}{1/n,\ldots ,1/n}}\},\;0\leq k<n. \end{array}$$Let $G_{0,a}=U_{0,n}=\left\{1\right\}$ and
$$\begin{array}{ccc} {\hat{\partial }}_{k}^{n}p & = & \frac{1}{1-p_{k}^{2^{n}}}{\left(\partial_{k}^{n}p\right)}_{\hat{k}}\;,\;p_{k}<1,n\in N,\\ {\hat{\partial }}_{k}^{0}p & = & p_{\hat{k}}/\left(1-p_{k}\right),\;p_{k}<1. \end{array}$$We denote by $\dot{=}$ equality up to inserting zeros and reordering.

**Lemma** **5.**
*Let p∈P and $\partial_{k}^{\infty }p:=\lim_{n\to \infty }\partial_{k}^{n}p$. Then,*

(17)
$$\begin{array}{ccc} \partial_{k}^{\infty }p & \dot{=} & g_{p_{k}}\times {\hat{\partial }}_{k}^{0}p,\;p_{k}<1, \end{array}$$


(18)
$$\begin{array}{ccc} \left\{1\right\} & \dot{=} & \{1\}\times \{1\}, \end{array}$$


(19)
$$\begin{array}{ccc} {\hat{\partial }}_{k}^{n}p & \dot{=} & g_{2^{n},p_{k}}\times {\hat{\partial }}_{k}^{0}p,\;p_{k}<1. \end{array}$$

*In particular, for a∈(0,1) and n∈N we have*

(20)
$$\begin{array}{ccc} \partial_{1}^{\infty }b_{a} & \dot{=} & g_{a}, \end{array}$$


(21)
$$\begin{array}{ccc} \partial_{1}^{\infty }g_{a} & \dot{=} & g_{1-a}\times g_{a}, \end{array}$$


(22)
$$\begin{array}{ccc} \partial_{1}^{\infty }G_{n,a} & \dot{=} & g_{a^{n}}\times g_{n,a}\;\dot{=}\;g_{a}\;\dot{=}\;G_{\infty ,a}, \end{array}$$


(23)
$$\begin{array}{ccc} \partial_{2}^{\infty }G_{n,a} & \dot{=} & g_{1-a}\times G_{n-1,a}, \end{array}$$


(24)
$$\begin{array}{ccc} \partial_{1}^{\infty }U_{k,n} & \dot{=} & g_{1-k/n}\times U_{k},\;1\leq k<n, \end{array}$$


(25)
$$\begin{array}{ccc} \partial_{2}^{\infty }U_{k,n} & \dot{=} & g_{1/n}\times U_{k-1,n-1},\;1\leq k<n, \end{array}$$


(26)
$$\begin{array}{ccc} \partial_{3}^{\infty }\partial_{2}b_{a} & \dot{=} & g_{a}\times b_{a}. \end{array}$$

*Furthermore, for a∈(0,1) and n∈N,*

(27)
$$\begin{array}{ccc} \partial_{1}G_{n,a} & \dot{=} & G_{2n,a}, \end{array}$$


(28)
$$\begin{array}{ccc} {\left(G_{n,a}\right)}_{\hat{2}} & = & aG_{n-1,a}. \end{array}$$



**Proof.** Without loss of generality, let k=1 in Equation (Equation 17). We have
$$\begin{array}{c} \partial_{1}^{n}p=\left\{p_{1}^{2^{n}}\right\}\cup \underset{\ell \in \{2^{n}-1,\ldots ,0\}}{⋃}p_{1}^{\ell }p_{\hat{1}}, \end{array}$$
since this is true for n=1 and
$$\begin{array}{ccc} \partial_{1}^{n+1}p & = & \partial_{1}\left(\partial_{1}^{n}p\right)=\partial_{1}\left(\left\{p_{1}^{2^{n}}\right\}\cup \underset{\ell \in \{2^{n}-1,\ldots ,0\}}{⋃}p_{1}^{\ell }p_{\hat{1}}\right)\\ & = & p_{1}^{2^{n}}\left(\left\{p_{1}^{2^{n}}\right\}\cup \underset{\ell \in \{2^{n}-1,\ldots ,0\}}{⋃}p_{1}^{\ell }p_{\hat{1}}\right)\cup \;\underset{\ell \in \{2^{n}-1,\ldots ,0\}}{⋃}p_{1}^{\ell }p_{\hat{1}}\\ & = & \left\{p_{1}^{2^{n+1}}\right\}\cup \underset{\ell \in \{2^{n+1}-1,\ldots ,2^{n}\}}{⋃}p_{1}^{\ell }p_{\hat{1}}\cup \underset{\ell \in \{2^{n}-1,\ldots ,0\}}{⋃}p_{1}^{\ell }p_{\hat{1}} \end{array}$$
by induction. Hence,
$$\begin{array}{c} \partial_{1}^{\infty }p\dot{=}\underset{\ell \in \{\ldots ,1,0\}}{⋃}p_{1}^{\ell }p_{\hat{1}}. \end{array}$$Now, $p_{1}^{\ell }p_{\hat{1}}=\left(1-p_{1}\right)p_{1}^{\ell }\cdot \frac{p_{\hat{1}}}{1-p_{1}}$ for $p_{1}<1$, implying Equation (Equation 17). Further, we have $\partial_{2}b_{a}=\left(1-a\right)b_{a}\cup \left\{a\right\}$, so that Equation (26) holds by Equation (Equation 17). The other equalities of the lemma are immediate. □

**Lemma** **6.***Let S:P→[0,∞] and f:[0,1]→[0,∞), such that Equation* (Equation 9) *holds true. Then,*(29)$$\begin{array}{ccc} S\left(\partial_{1}^{n}p\right) & = & \left(S\left(p\right)-1\right)F_{n}\left(p_{1}\right)+1,\;p\in P,n\in N, \end{array}$$(30)$$\begin{array}{ccc} S(G_{2n,a}) & = & \left(1+f\left(a^{n}\right)\right)S\left(G_{n,a}\right)-f\left(a^{n}\right),\;n\in N. \end{array}$$

**Proof.** It is easy to check by induction that Equation (Equation 9) implies
$$\begin{array}{c} S\left(\partial_{1}^{n}p\right)=S\left(p\right)\prod_{k=0}^{n-1}\left(1+f(p_{1}^{2^{k}})\right)-\sum_{\ell =0}^{n-1}f\left(p_{1}^{2^{\ell }}\right)\prod_{k=\ell +1}^{n-1}\left(1+f(p_{1}^{2^{k}})\right). \end{array}$$Now,
$$\begin{array}{c} 1+\sum_{i=1}^{n}c_{i}\prod_{j=i+1}^{n}\left(1+c_{j}\right)=\prod_{j=1}^{n}\left(1+c_{j}\right),\;c_{j}\in R,n\in N, \end{array}$$
by induction, so that the first assertion of the lemma holds. Equation (Equation 27) yields the second assertion. □

### 3.3. Proof of Theorem 1

Assume that *H* is the Shannon entropy. Then,
$$H\left(g_{a}\right)=-\sum_{k=0}^{\infty }\left(1-a\right)a^{k}\log\left(\left(1-a\right)a^{k}\right)=-\left(a\log a+\left(1-a\right)\log\left(1-a\right)\right)/\left(1-a\right)\in \left(0,\infty \right)$$
for a∈(0,1). All the other assertions in the second condition hold obviously, when *f* is chosen to be the identity. Now, assume that the second condition applies. By Condition 2(a), Condition 2(b) holds for all p∈P. Condition (f) and Equation (18) yield H({1})=0. It is easy to check that $H\left(\partial_{1}^{n}p\right)=H\left(p\right)F_{n}\left(p_{1}\right),$ so that
(31)$$\begin{array}{c} H\left(\partial_{1}^{\infty }p\right)=H\left(\lim_{n\to \infty }\partial_{1}^{n}p\right)=\lim_{n\to \infty }H\left(\partial_{1}^{n}p\right)=H\left(p\right)F\left(p_{1}\right). \end{array}$$The function *F* is finite due to Conditions 2(e) and 2(f) and Equations (21) and (Equation 31), i.e.,
$$\begin{array}{c} H\left(g_{a}\right)F\left(1-a\right)=H\left(g_{1-a}\right)+H\left(g_{a}\right). \end{array}$$It follows that
$$F\left(1-a\right)-1=\frac{H(g_{1-a})}{H(g_{a})},\;a\in \left(0,1\right).$$Replacing *a* by 1−a in the above equation yields
(32)$$\begin{array}{c} (F(1-a)-1)(F(a)-1)=1,\;a\in (0,1). \end{array}$$Now,
(33)$$\begin{array}{c} H\left(b_{a}\right)F\left(a\right)=H\left(g_{a}\right)>0, \end{array}$$
by Equations (Equation 20) and (Equation 31) and Condition 2(e). Equations (26), (Equation 8), and (Equation 31) yield
$$\begin{array}{c} H\left(b_{a}\right)\left(1+f\left(1-a\right)\right)F\left(a\right)=H\left(g_{a}\right)+H\left(b_{a}\right); \end{array}$$
thus, f(1−a)F(a)=1 for a∈(0,1). Equation (Equation 32) now writes f(a)+f(1−a)=1. Plugging in $p=b_{a}$ into Equation (Equation 8) shows that *f* is continuous on (0,1). Now, Lemma 1 yields f(a)=a and F(a)=1/(1−a) for a∈(0,1). Equation (Equation 8) implies that f(0)=0, so that *f* is the identity on [0,1). Lemma 3 delivers $H(U_{n})=c\log n$ for a c≥0. Condition 2(f), Equations (Equation 17) and (Equation 31) yield
(34)$$\begin{array}{c} H\left(p\right)F\left(p_{1}\right)=H\left(g_{p_{1}}\right)+H\left({\hat{\partial }}_{1}^{0}p\right),\;p_{1}<1. \end{array}$$This implies that
$$\begin{array}{c} c\log n\cdot F\left(1/n\right)=H\left(g_{1/n}\right)+c\log\left(n-1\right) \end{array}$$
for n≥2, so that c>0 and
$$\begin{array}{c} c^{-1}H\left(g_{1/n}\right)=\frac{1}{1-1/n}\log n-\log\left(n-1\right)=-\frac{1/n}{1-1/n}\log\frac{1}{n}-\log\left(1-\frac{1}{n}\right). \end{array}$$Henceforth, we may assume that c=1. Due to the recursion Formula (Equation 34), it suffices to show that the above formula extends to $H\left(g_{a}\right)=-a{\left(1-a\right)}^{-1}\log a-\log\left(1-a\right)$, a∈(0,1), to finish the proof. By Equations (25) and (Equation 31), we obtain
$$H\left(U_{n-k,n}\right)F\left(1/n\right)=H\left(g_{1/n}\right)+H\left(U_{n-k-1,n-1}\right),\;k\geq 1,$$
i.e.,
$$H\left(U_{n-k,n}\right)=\sum_{i=0}^{n-k-1}\frac{H(g_{1/(n-i)})}{\prod_{j=n-i}^{n}F\left(1/j\right)}=\frac{1}{n}\sum_{i=0}^{n-k-1}\left(n-i-1\right)H\left(g_{1/(n-i)}\right)$$
by iteration. On the other hand, by Equations (24) and (Equation 31),
$$H\left(U_{n-k,n}\right)F\left(k/n\right)=H\left(g_{k/n}\right)+\log\left(n-k\right),$$
so that
$$\begin{array}{ccc} H\left(g_{k/n}\right)+\log\left(n-k\right) & = & \frac{1}{n-k}\sum_{i=0}^{n-k-1}\left(\left(n-i\right)\log\left(n-i\right)-\left(n-i-1\right)\log\left(n-i-1\right)\right)\\ & = & \frac{1}{n-k}\left(n\log n-k\log k\right)=-\frac{k/n}{1-k/n}\log\frac{k}{n}+\log n. \end{array}$$Condition 2(a), i.e., the continuity of *H*, finalizes the proof.

### 3.4. Proof of Theorem 2

Assume that *H* is the Rényi entropy. Let S(p)=exp((1−α)H(p)), i.e., $S\left(p\right)={∥p∥}_{\alpha }^{\alpha }$. Then,
$$S\left(\partial_{1}p\right)=p_{1}^{\alpha }\sum_{i}p_{i}^{\alpha }+\sum_{i}p_{i}^{\alpha }-p_{1}^{\alpha },$$
so that $f\left(r\right)=r^{\alpha }$. Reversely, assume that the second condition holds. Since H({1})=0 by Condition 2(f) and Equation (18), we have S({1})=1. Furthermore, *S* is continuous. If S(p)=1, then $1=S\left(\partial_{1}p\right)=S\left(\partial_{2}p\right)=\ldots =S\left(\partial_{1}^{\infty }p\right)$ by Equation (Equation 9). Hence, $S(b_{a})\neq 1$ for a∈(0,1), as otherwise we would obtain a contradiction to Condition 2(e) by Equation (Equation 20). It follows that $f\left(a\right)=\left(S\left(\partial_{1}b_{a}\right)-S\left(b_{a}\right)\right)/\left(S\left(b_{a}\right)-1\right)$ is continuous on (0,1). Equation (Equation 9) implies that f(0)(S({0}∪p)−1)=0 for all p∈P, so that f(0)=0. Condition 2(f) reads
(35)$$\begin{array}{c} S(p\times q)=S(p)S(q). \end{array}$$By Condition 2(a), Condition 2(b) holds for all p∈P. Henceforth, we assume that a∈(0,1). Condition 2(e), Equations (Equation 29) and (21) yield
$$\begin{array}{c} \infty >S\left(g_{a}\right)S\left(g_{1-a}\right)\leftarrow \left(S\left(g_{1-a}\right)-1\right)F_{n}\left(a\right)+1, \end{array}$$
as n→∞. Hence, $F\left(a\right)=\lim_{n\to \infty }F\left(a\right)$ is finite. Now, by Equations (Equation 17), (Equation 35) and (Equation 29),
(36)$$\begin{array}{c} \left(S\left(p\right)-1\right)F\left(p_{1}\right)+1=S\left(g_{p_{1}}\right)S\left({\hat{\partial }}_{1}^{0}p\right),\;p_{1}<1. \end{array}$$In particular,
(37)$$\begin{array}{c} \left(S\left(b_{a}\right)-1\right)F\left(a\right)+1=S\left(g_{a}\right). \end{array}$$We obtain by means of Equation (26) that
$$\left[S\left(b_{a}\right)\left(1+f\left(1-a\right)\right)-f\left(1-a\right)-1\right]F\left(a\right)+1=S\left(g_{a}\right)S\left(b_{a}\right),$$
so that, by Equation (Equation 37),
$$\left[S\left(b_{a}\right)\left(1+f\left(1-a\right)\right)-f\left(1-a\right)\right]F\left(a\right)-S\left(b_{a}\right)F\left(a\right)+S\left(g_{a}\right)=S\left(g_{a}\right)S\left(b_{a}\right),$$
i.e., as $S(b_{a})\neq 1$,
(38)$$\begin{array}{c} f\left(1-a\right)F\left(a\right)=S\left(g_{a}\right). \end{array}$$Since $S(g_{a})\neq 0$, it follows that f(a),F(a)≠0. Equations (Equation 36) and (Equation 38) yield for $p=g_{1-a}$ that $1-F{\left(a\right)}^{-1}=\left(1-f\left(1-a\right)\right)S\left(g_{1-a}\right)$, so that Equation (Equation 36) now reads
(39)$$\begin{array}{c} S\left(p\right)-\left(1-f\left(1-p_{1}\right)\right)S\left(g_{1-p_{1}}\right)=f\left(1-p_{1}\right)S\left({\hat{\partial }}_{1}^{0}p\right), \end{array}$$
again by Equation (Equation 38). In line with the idea from Equation (23), Equations (Equation 39) and (28) yield
(40)$$\begin{array}{ccc} S(G_{n,a}) & = & \left(1-f\left(a\right)\right)S\left(g_{a}\right)+f\left(a\right)S\left(G_{n-1,a}\right)\\ & = & \left(1-f\left(a\right)\right)S\left(g_{a}\right)\sum_{k=0}^{n-1}f^{k}\left(a\right)+f^{n}\left(a\right)\\ & = & f^{n}\left(a\right)+\left(1-f^{n}\left(a\right)\right)S\left(g_{a}\right). \end{array}$$Plugging in Equation (Equation 40) into Equation (30) yields
$$\begin{array}{c} f^{2n}\left(a\right)+\left(1-f^{2n}\left(a\right)\right)S\left(g_{a}\right)=\left(1+f\left(a^{n}\right)\right)\left(f^{n}\left(a\right)+\left(1-f^{n}\left(a\right)\right)S\left(g_{a}\right)\right)-f\left(a^{n}\right), \end{array}$$
i.e.,
$$\begin{array}{c} \left(f\left(a^{n}\right)-f^{n}\left(a\right)\right)\left(1-f^{n}\left(a\right)\right)\left(S\left(g_{a}\right)-1\right)=0. \end{array}$$Assume that f(a)=1. Then, by Equation (Equation 40), $1=S\left(G_{n,a}\right)\to S\left(g_{a}\right)$ as n→∞. This contradicts $S(g_{a})\neq 1$. Hence, we obtain for n=2 that
(41)$$\begin{array}{c} f^{2}\left(a\right)=f\left(a^{2}\right),\;a\in \left(0,1\right). \end{array}$$Lemma 3 implies that $H\left(U_{n}\right)=\tilde{c}\log n$ for a $\tilde{c}\geq 0$; hence, $S\left(U_{n}\right)=n^{c}$ for $c=\gamma \tilde{c}\in R$. By $S\left(U_{2}\right)=S\left(b_{1/2}\right)\neq 1$, it follows that c≠0. Equation (Equation 39) gives
(42)$$\begin{array}{c} \left(1-f\left(1-1/n\right)\right)S\left(g_{1-1/n}\right)=n^{c}-f\left(1-1/n\right){\left(n-1\right)}^{c}. \end{array}$$Furthermore, taking the ideas from Equations (24) and (25), we obtain, by Equation (Equation 39),
$$S\left(U_{k,n}\right)=\left(1-f\left(k/n\right)\right)S\left(g_{k/n}\right)+f\left(k/n\right)k^{c}$$
and
$$\begin{array}{ccc} S(U_{k,n}) & = & \left(1-f\left(1-1/n\right)\right)S\left(g_{1-1/n}\right)+f\left(1-1/n\right)S\left(U_{k-1,n-1}\right)\\ & = & n^{c}+f\left(1-1/n\right)\left[-{\left(n-1\right)}^{c}+S\left(U_{k-1,n-1}\right)\right]\\ & = & n^{c}+\left(1-{\left(n-k\right)}^{c}\right)\prod_{j=0}^{k-1}f\left(1-1/\left(n-j\right)\right), \end{array}$$
by Equation (Equation 42) and iteration. Thus, for 0<k<n, we have
(43)$$\begin{array}{c} \left(1-f\left(k/n\right)\right)S\left(g_{k/n}\right)=n^{c}+\left(1-{\left(n-k\right)}^{c}\right)\frac{M_{n}}{M_{n-k}}-f\left(k/n\right)k^{c}, \end{array}$$
where
$$M_{1}=1,\;M_{n}=\prod_{j=2}^{n}f\left(1-1/j\right),\;n\geq 2.$$Note that $M_{n}>0$ for all n∈N as a consequence of Equation (Equation 38). We obtain for n=2k in Equation (Equation 43) that
(44)$$\begin{array}{c} \left(1-f\left(1/2\right)\right)S\left(g_{1/2}\right)={\left(2k\right)}^{c}+\left(1-k^{c}\right)\frac{M_{2k}}{M_{k}}-f\left(1/2\right)k^{c}. \end{array}$$For k=1, this is
(45)$$\begin{array}{c} \left(1-f\left(1/2\right)\right)S\left(g_{1/2}\right)=2^{c}-f\left(1/2\right). \end{array}$$Plugging Equation (Equation 45) into Equation (Equation 44) we obtain $M_{2k}/M_{k}=2^{c}-f\left(1/2\right)$ for k>1. An immediate consequence of this constant ratio is that $M_{2k}/M_{2\ell }=M_{k}/M_{\ell }$ for ℓ,k>1. Equation (Equation 43), the symmetry of the Bernoulli distribution, and the fact that, by Equation (Equation 39), $S\left(b_{a}\right)=\left(1-f\left(a\right)\right)S\left(g_{a}\right)+f\left(a\right)$, yield
$$\begin{array}{c} \left(1-{\left(n-k\right)}^{c}\right)\frac{M_{n}}{M_{n-k}}-f\left(k/n\right)\left(k^{c}-1\right)=\left(1-k^{c}\right)\frac{M_{n}}{M_{k}}-f\left(1-k/n\right)\left({\left(n-k\right)}^{c}-1\right), \end{array}$$
i.e., for all ℓ≥0 and 1<k<n−1,
(46)$$\begin{array}{ccc} f(k/n) & = & \frac{M_{2^{\ell }n}}{M_{2^{\ell }k}}-N\left(2^{\ell }\right)\left[\frac{M_{2^{\ell }n}}{M_{2^{\ell }\left(n-k\right)}}-f\left(1-k/n\right)\right]\\ & = & \frac{M_{n}}{M_{k}}-N\left(2^{\ell }\right)\left[\frac{M_{n}}{M_{n-k}}-f\left(1-k/n\right)\right] \end{array}$$
with
$$N\left(j\right)=\frac{1-j^{c}{\left(n-k\right)}^{c}}{1-{\left(jk\right)}^{c}}.$$In order for the right-hand side of Equation (Equation 46) to be independent of *ℓ*, we necessarily have $f\left(1-k/n\right)=M_{n}/M_{n-k}$. Now, Equation (Equation 43) ensures that for all ℓ≥1 and 1<k<n−1, we have
$$\begin{array}{ccc} \left(1-f\left(k/n\right)\right)S\left(g_{k/n}\right) & = & \ell^{c}n^{c}+\left(1-\ell^{c}{\left(n-k\right)}^{c}\right)f\left(1-k/n\right)-f\left(k/n\right)\ell^{c}k^{c}\\ & = & f\left(1-k/n\right)+k^{c}l^{c}\left({\left(n/k\right)}^{c}-{\left(n/k-1\right)}^{c}f\left(1-k/n\right)-f\left(k/n\right)\right). \end{array}$$In order for the right-hand side to be independent of *ℓ*, we necessarily have
$$\begin{array}{c} {\left(k/n\right)}^{c}f\left(k/n\right)+{\left(1-k/n\right)}^{c}f\left(1-k/n\right)=1. \end{array}$$Since *f* is continuous, we even have
$$a^{c}f\left(a\right)+{\left(1-a\right)}^{c}f\left(1-a\right)=1,\;a\in \left(0,1\right).$$Let $\tilde{f}\left(a\right)=a^{c}f\left(a\right)$; then $\tilde{f}\left(a^{2}\right)={\tilde{f}}^{2}\left(a\right)$ by Equation (Equation 41) and $\tilde{f}\left(a\right)+\tilde{f}\left(1-a\right)=1$. Lemma 1 yields that $\tilde{f}$ is the identity on (0,1); hence, $f\left(a\right)=a^{\alpha }$ with α=1−c.

### 3.5. Proof of Theorem 3

Assume that *H* is the min-entropy and let *f* be the identity. Then, Equation (Equation 10) obviously holds true. Reversely, assume that the second condition of the theorem applies. Condition 2(f) and Equation (18) imply that H({1})=0 and S({1})=1. Equations (Equation 10) and (Equation 17) yield
(47)$$\begin{array}{c} S\left(p\right)=S\left(\partial_{1}^{\infty }p\right)\vee \Phi \left(p_{1}\right)=S\left(g_{p_{1}}\right)S\left({\hat{\partial }}_{1}^{0}p\right)\vee \Phi \left(p_{1}\right),\;p_{1}<1, \end{array}$$
where $\Phi \left(a\right)=\sup_{k=0}^{\infty }f\left(a^{2^{k}}\right)$. Equations (Equation 10), (Equation 17), and (21) deliver $S\left(g_{a}\right)=S\left(g_{1-a}\right)S\left(g_{a}\right)\vee \Phi \left(1-a\right)$, so that Φ(a) is finite for a∈(0,1). Since $S\left(g_{1-a}\right)\in \left(0,1\right)$ by Condition 2(e), it follows that
(48)$$\begin{array}{c} S\left(g_{a}\right)=\Phi \left(1-a\right)\in \left(0,1\right),\;a\in \left(0,1\right). \end{array}$$Thus, the recursion formula (Equation 47) determines *H* uniquely as soon as Φ is determined. Since *S* is continuous, the function Φ is continuous on (0,1). Lemma 3 yields
$$\begin{array}{c} n^{-c}=S\left(U_{n}\right)=\Phi \left(1-1/n\right){\left(n-1\right)}^{-c}\vee \Phi \left(1/n\right),\;n\in N, \end{array}$$
for some c≥0. In particular, $\Phi \left(1/n\right)\leq n^{-c}$ and $\Phi \left(1/2\right)=2^{-c}$. Assuming c=0 leads to 1=Φ(1−1/n)∨Φ(1/n), contradicting Equation (Equation 48). Equation (Equation 10) implies that $0\leq \Phi \left(0\right)\leq \inf_{n\in N}S\left(U_{n}\right)=0$. Equations (Equation 47), (22), and (Equation 48) yield
$$\begin{array}{c} S\left(G_{n,a}\right)=S\left(g_{a}\right)\vee \Phi \left(a^{n}\right)=\Phi \left(1-a\right)\vee \Phi \left(a^{n}\right). \end{array}$$On the other hand, Equations (23) and (Equation 48) and iterating Equation (Equation 47) yield
$$\begin{array}{c} S\left(G_{n,a}\right)=\Phi \left(a\right)S\left(G_{n-1,a}\right)\vee \Phi \left(1-a\right)=\Phi^{n}\left(a\right)\vee \Phi \left(1-a\right). \end{array}$$We define $U_{n,n}=\left\{0\right\}\cup U_{n}$ and Φ(1)=1. Then, Equations (24) and (25) yield
$$\begin{array}{c} S\left(U_{n-k,n}\right)=\Phi \left(1-k/n\right){\left(n-k\right)}^{-c}\vee \Phi \left(k/n\right) \end{array}$$
and
$$\begin{array}{ccc} S(U_{n-k,n}) & = & \Phi \left(1-1/n\right)S\left(U_{n-k-1,n-1}\right)\vee \Phi \left(1/n\right)\\ & = & \prod_{m=k+1}^{n}\Phi \left(1-1/m\right)\vee \overset{n}{\underset{\ell =k+1}{⋁}}\Phi \left(1/\ell \right)\prod_{m=\ell +1}^{n}\Phi \left(1-1/m\right) \end{array}$$
for 0≤k<n. Lemma 2 finalizes the proof.

### 3.6. Proof of Theorem 4

Assume that *H* is the Hartley entropy and let $f\left(a\right)=1_{\left(0,1\right]}\left(a\right)$. Then, the second condition in the theorem holds true. Reversely, assume that the second condition applies. Equations (Equation 12) and (19) yield
$$S\left(g_{2^{n},p_{1}}\right)S\left({\hat{\partial }}_{1}^{0}p\right)=S\left({\hat{\partial }}_{1}^{n}p\right)=S\left({\hat{\partial }}_{1}^{0}p\right)F_{n}\left(p_{1}\right),\;p_{1}<1,p\in P,$$
so that
(49)$$\begin{array}{c} S\left(g_{2^{n},a}\right)=F_{n}\left(a\right),\;a\in \left(0,1\right), \end{array}$$
as S(p)∈[1,∞) for $p\in P_{0}$. Since $F_{n}\left(a\right)$ is monotone in *n*, the limit $F\left(a\right)=\lim_{n\to \infty }F_{n}\left(a\right)$ always exists, i.e., F(a)∈[1,∞]. Assume that $F(a_{*})<\infty$ for some $a_{*}\in \left(0,1\right)$. Condition 2(f) and Equations (Equation 29), (Equation 17), (Equation 49), and (Equation 11) yield for $p\in P_{0}$ that
(50)$$\begin{array}{ccc} \infty & > & \left(S(\left\{a_{*}\right\}\cup \left(1-a_{*}\right)p)-1\right)F\left(a_{*}\right)+1\\ & = & \lim_{n\to \infty }S\left(\partial_{1}^{n}\left(\left\{a_{*}\right\}\cup \left(1-a_{*}\right)p\right)\right)\\ & = & \lim_{n\to \infty }\frac{S(a_{*}^{2^{n}}\left\{1\right\}\cup \left(1-a_{*}^{2^{n}}\right)g_{2^{n},a_{*}}\times p)}{S(g_{2^{n},a_{*}}\times p)}\lim_{n\to \infty }F_{n}\left(a_{*}\right)S\left(p\right)\\ & = & F\left(a_{*}\right)S\left(p\right). \end{array}$$Assume that $F(a_{*})=1$. Equations (Equation 50) and (Equation 11) deliver for $p\in P_{0}$
(51)$$\begin{array}{ccc} \infty >S(p) & = & S\left(\left\{a_{*}\right\}\cup \left(1-a_{*}\right)p\right)=\ldots =S\left(\left(1-{\left(1-a_{*}\right)}^{n}\right)g_{n,1-a_{*}}\cup {\left(1-a_{*}\right)}^{n}p\right)\\ & = & \lim_{n\to \infty }S\left(g_{n,1-a_{*}}\right). \end{array}$$Lemma 3 yields $S\left(U_{n}\right)=n^{c}$ for c≥0. Now, Equation (Equation 51) implies that c=0, while the assumption in the theorem, $H(U_{2})>0$, implies that c>0. Hence, $F(a_{*})$ cannot be equal to 1. Plugging $p=G_{k,1-a_{*}}$ into Equation (Equation 50) gives
$$S\left(G_{k-1,1-a_{*}}\right)F\left(a_{*}\right)=\lim_{n\to \infty }S\left(\partial_{2}^{n}G_{k,1-a_{*}}\right)=\left(S\left(G_{k,1-a_{*}}\right)-1\right)F\left(a_{*}\right)+1,$$
by Equation (28). As F≥1, iterating yields $S\left(G_{k,1-a_{*}}\right)=1+k\left(1-1/F\left(a_{*}\right)\right)$, so that
$$\begin{array}{c} 1+2k\left(1-1/F\left(a_{*}\right)\right)=\left(1+f\left(a_{*}^{k}\right)\right)\left(1+k\left(1-1/F\left(a_{*}\right)\right)\right)-f\left(a_{*}^{k}\right),\;k\in N, \end{array}$$
by Equation (30). Since $F(a_{*})\neq 1$, we obtain $f(a_{*}^{k})=1$, k∈N, contradicting the finiteness of $F(a_{*})$. Hence, F(a)=∞ for all a∈(0,1). Equations (Equation 29), (Equation 11), and (Equation 49) yield
$$S\left(p\right)-1=\frac{S(\partial_{1}^{n}p)-1}{F_{n}\left(p_{1}\right)S\left({\hat{\partial }}_{1}^{0}p\right)}S\left({\hat{\partial }}_{1}^{0}p\right)\to S\left({\hat{\partial }}_{1}^{0}p\right),\;p_{1}\in \left(0,1\right),S\left(p\right)<\infty ,$$
i.e.,
(52)$$\begin{array}{c} S\left(p\right)=S\left({\hat{\partial }}_{1}^{0}p\right)+1,\;p_{1}\in \left(0,1\right),S\left(p\right)<\infty . \end{array}$$The assertion of the theorem follows for $p\in P_{0}$ by Conditions 2(b)–2(d) and the fact that S({1})=1. In particular, $f\left(a\right)=1_{\left(0,1\right]}\left(a\right)$. Assume now that S(p)<∞ for some $p\in P\setminus P_{0}$. Without loss of generality, $p_{i}>0$ for all i∈N. Then, by induction, Equation (Equation 52) delivers $S({\left({\hat{\partial }}_{1}^{0}\right)}^{n}p)<\infty$ and $S\left(p\right)=S({\left({\hat{\partial }}_{1}^{0}\right)}^{n}p)+n$, n∈N. This contradicts S≥1 and S(p)<∞. Hence, S(p)=∞ for $p\in P\setminus P_{0}$.

## 4. Discussion

Although all conditions in the characterizations of Shannon’s and Rényi’s entropy have intrinsic motivations and interpretations, their relevance might differ. Continuity and finiteness offer advantages also from a technical point of view and will usually be retained. Condition 2(b) is intrinsic for probability distributions. The additivity might be considered to be axiomatic. Therefore, we would like to discuss here the importance of the following three points:The discrete nature of the objects;The hierarchical, self-similar structure of the objects, cf. Condition 2(g);The parity among the objects at each hierarchical level, cf. Condition 2(b).

### 4.1. Practical Implications

The three conditions above are obviously tailormade for tree-like graphs and other hierarchical structures, without the necessity to appeal information-theorical arguments. Examples that may fit our approach are, for instance,

Classification systems: forms of art (e.g., music genres), languages and their dialects.Biology: taxonomie, diversity, genome.Administration units: districts, corporate structure, military.Universe: clustering.Material sciences [18,19].Networks [20].Statistics: graphs and cluster analysis.

For all these examples, Shannon’s or Rényi’s entropy may at least be a default choice. The min-entropy may suit graphical models for extremes [21].

### 4.2. Modifications

The three properties above might be the most questionable from a nonideal, practical point of view. While we believe that nondiscrete objects lead to different theories, modifications and variants of the self-similarity property and the parity property might be accessible to some extent by theoretical advancements of this framework. Also modifications of the operator *∂* could be of practical interest. Since the tree built by $\partial^{\infty }$ can be seen as a deterministic two-type Kesten tree [22], randomizations of $\partial^{\infty }$ bridge to the Galton–Watson processes.

### 4.3. The Geometric Distribution

In comparison to Faddeev’s approach, our proofs emphasize the role of the geometric distribution $g_{a}$, which arises as the infinite iteration of the Bernoulli distribution $b_{a}$, a∈(0,1), i.e., $\partial_{1}^{\infty }b_{a}=g_{a}$. The uniqueness proof for the Shannon entropy uses almost only the resulting proportionality relation of the geometric distribution, simplifying Faddeev’s approach. The important role of the geometric distribution is already known in information theory: it maximizes the Shannon entropy among discrete probability distributions given the mean ([23], Theorem 5.8) and minimizes the min-entropy under fixed variance, among all discrete log-concave random variables in Z ([12], Theorem 1). The geometrical distribution also appears in branching processes [24] and in physics to describe the canonical ensemble [25]. By our research, we add that the geometric distribution is central to describe the self-similarity law of the Shannon entropy. Obviously, the operator *∂* can be inverted, so that the distribution is not refined at a leaf of the graph, but the structure is extended at the root, i.e., $\partial^{-1}\left(q,a,p\right):=\left(1-a\right)p\cup aq$, so that $\partial^{-1}\left(\partial_{1}^{\infty }p,p_{1},{\hat{\partial }}_{1}^{0}p\right)=\partial_{1}^{\infty }p$. Hence, $\partial_{1}^{\infty }p$ can be interpreted as a kind of limit distribution that is invariant to $\partial^{-1}$. Whether this leads to any formulation of a limit law including a domain of attraction is an open question. Note that the operator $\partial^{-1}$ has already been addressed in [8].

### 4.4. On the Proportionality Relation Characterizing the
Rényi Entropy

Although the characterization and the proofs of the Shannon and the Rényi entropy follow the same scheme, they show remarkable differences. For instance, the generalization $G_{n,a}$ of the geometric distribution (see Equation (Equation 15)) does not play a role in the case of the Shannon entropy. Also, the functional characterization in Equation (Equation 8) of the Shannon entropy is not the limit of the characterization of the Rényi entropy (see Equation (Equation 9)). However, Equation (Equation 8) might be understood as the derivative of Equation (Equation 9) in the following sense, cf. [26]. Assume that *f* in Equation (Equation 9) is differentiable. Then,
$$\frac{d}{d\alpha }S\left(\partial_{1}p\right)=\frac{d}{d\alpha }\left(\left(1+f\left(p_{1}\right)\right)S\left(p\right)-f\left(p_{1}\right)\right).$$For α=1, we define $S\left(p\right)=\lim_{\alpha \to 1}{∥p∥}_{\alpha }^{\alpha }$. Then, we obtain
$$\frac{d}{d\alpha }S\left(\partial_{1}p\right)=\left(1+f\left(p_{1}\right)\right)\frac{d}{d\alpha }S\left(p\right).$$

Another interpretation of our results is that the order-α information energy *S* is not a natural transformation of the Rényi entropy, in contrast to
T(p)=S(p)−S({1}),
which equals the Harvda–Charvát entropy [27] or Tsallis entropy [28], up to a multiplicative constant. Therefore, *T* bridges to the generalized information function ([5], Section 6.2), but also to the definite functions ([29], Chapter 3, Corollary 3.3 and proof of Theorem 2.2, and Chapter 6, Example 5.16). Indeed, Equation (Equation 9) can be rewritten in full analogy to Equation (Equation 8), namely, as
$$\begin{array}{c} T\left(\partial_{1}p\right)=\left(1+f\left(p_{1}\right)\right)T\left(p\right), \end{array}$$
where $T\left(\partial_{1}p\right)=S\left(\partial_{1}p\right)-S\left(\left\{1\right\}\right)$. Note that also the interpretation $T\left(\partial_{1}p\right)=S\left(\partial_{1}p\right)-S\left(\partial_{1}\left\{1\right\}\right)$ is possible.

### 4.5. Tsallis and Other Entropies

It is an open question whether further practically relevant chain-rule-like conditions such as Condition 2(g) can be found, or, equivalently, recursive formulae such as Equations (Equation 34), (Equation 36), and (Equation 47), and, if so, whether they will be based on the geometric distribution. Reversely, it is also unclear which other existing definitions of an entropy can be described by conditions similar to Conditions 2(a)–2(g). Reference [15] considers additivity, i.e., Condition 2(f), as the only axiomatic property that an entropy must have. The prevalent interpretation of the Tsallis entropy [28] is that it is not additive; hence, the Tsallis entropy and many others are, per se, excluded from this framework, or the framework would have to be largely extended. Reference [15] remarks that the additivity law has two ingredients—the independence of the components before being joined and the requirement that the components do not interact when being joined. In other words, one may add an additivity law to the Tsallis entropy so that the Tsallis entropy indeed fits this framework. The practical use of such an approach may show the future.

### 4.6. Shannon’s Notion of Entropy

Shannon [30] took a very general perspective on which properties an entropy should have, namely, that “any monotonic function of this number [the finite number of messages] can be regarded as a measure of the information produced when one message is chosen from the set, all choices being equally likely”. Further, Shannon justified his and Hartley’s choice by the statement that “the logarithmic measure is more convenient”. Shannon added three arguments, namely, “it is practically more useful”, “it is nearer to our intuitive feeling as to the proper measure”, and “it is mathematically more suitable”. Shannon is known for his repeated warnings against the blind use of his approach outside the framework of telecommunication [31]. The approach of [15] is, in this spirit, trying to make Shannon’s three arguments mathematically more concrete and generally applicable. To this end, the entropy is characterized in algebraic terms only, which reflects the desired properties. Ideally, these properties are rich enough such that the choice of entropy becomes unique. Preferably, the desired properties stem from common sense, invariance considerations, and limit laws. This paper and the presented characterizations of the entropies follow these requirements, giving a deep justification for the use of the Shannon entropy in a large spectrum of applications. But it also indicates its limits of application, unless further deep characterizations are found.

## 5. Conclusions

The characterizations given here are not only novel with respect to splitting the chain rule into its natural components, they are also novel in the sense that only a rather weak relation is required for the chain rule, i.e., the proportionality factor itself is not given, but arises naturally. Furthermore, the recursion formulae have the advantage that they allow an interpretation as an intrinsic property, since only a single distribution is involved. Further, the recursion formulae may allow additional practical interpretations such as self-similarity or scale invariance. Lastly, while Equation (Equation 2) is a rule, proportionality can be considered a law. This yields a deeper understanding of the Shannon entropy and may justify a broad spectrum of applications outside telecommunication.

## Data Availability

No new data were created or analyzed in this study. Data sharing is not applicable to this article.

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
