# Peer review of "An Intrinsic Characterization of Shannon’s and Rényi’s Entropy"

_entropy, 2024, doi:10.3390/e26121051_

Round 1
Reviewer 1 Report
Comments and Suggestions for Authors
In the study entitled “An intrinsic characterization of Shannon’s and Rényi’s entropy”, the Authors attempt to show that the chain rule, familiar in the Shannon’s entropy, can be split into two natural components. As a results, the variant of the chain rule, that involves only a single elementary distribution, is shown to be a proportionality relation, allowing vague interpretation as self-similarity (intrinsic property of the Shannon entropy).
The presented paper appears to be free of a major error and written in alignment with the best practices of presenting scientific results. For most of the time, the formal side of the manuscript is top notch. As for the substantive aspects of the article, the presented results have fundamental character and may be of interested to the scientific community. My general assessment of the presented study is mostly positive I would like to recommend this paper for publication.
Reviewer 2 Report
Comments and Suggestions for Authors
The manuscript offers a novel perspective on Shannon's and Rényi's entropy by providing an intrinsic characterization that decomposes the chain rule into two natural components: the additivity property and a proportionality relation. It highlights the self-similarity of entropy as a fundamental property and extends this approach to Rényi entropy and min-entropy. This work presents a fresh interpretation of entropy with potential applications beyond traditional domains, such as telecommunications and statistical analysis. It also deepens the conceptual understanding of entropy and suggests broader applicability to recursive systems and scale-invariant processes.
The references are relevant and include key foundational works in entropy and information theory, with most citations being recent or classic contributions. The decomposition of the chain rule and its interpretation as a self-similar property represent an original and significant contribution. The proofs are rigorous, though some technical assumptions may be restrictive. While the manuscript is well-written, additional examples and clarifications could enhance its accessibility and broaden its appeal to a wider audience.
Several questions arise that could enhance the clarity and impact of the paper:
1. Characterization of Rényi Entropy (Section 2, Theorem 2): Can the technical assumptions, such as the requirement for real analyticity, be relaxed without compromising the core results?
2. Applications of Geometric Distribution (Section 2.2): Can the role of the geometric distribution as a limit of Bernoulli distributions be directly linked to practical systems, such as models of information loss or hierarchical data structures?
3. Extension to Dynamic Systems: Could the proposed interpretation of proportionality and self-similarity in entropy be extended to time-varying probabilistic systems, such as those in temporal data analysis or adaptive models?
4. Comparison with Tsallis Entropy (Lines 232–248): How does the proposed framework align with other entropy formulations, such as Tsallis entropy? Could it also fit within the characterizations described in this manuscript?
We recommend accepting the paper after addressing these minor revisions to enhance its clarity, accessibility, and practical relevance.
Round 2
Reviewer 1 Report
Comments and Suggestions for Authors
As previously I recommend this study for publication.
Reviewer 2 Report
Comments and Suggestions for Authors
I have reviewed the revised manuscript titled "An Intrinsic Characterization of Shannon’s and Rényi’s Entropy" by M. Schlather and C. Ditscheid, and I am pleased to recommend its acceptance for publication in its current form.
The authors have addressed the points raised during the review process thoroughly and effectively. The restructuring of the manuscript, the inclusion of a new “Discussion” section, and the clarification of shared results with a new lemma significantly enhance the paper's readability and accessibility. The removal of the assumption of real analyticity further strengthens the core results and aligns the characterization of entropy across various frameworks, including Shannon, Rényi, and min-entropy.
The expanded discussion of practical examples and potential applications broadens the relevance of the work and highlights its impact on areas such as telecommunications, statistical analysis, and recursive systems. While the manuscript leaves some future research directions open, such as extending the framework to dynamic systems, this is a natural aspect of exploratory research and provides a solid foundation for further studies.
Overall, this paper provides a novel perspective on entropy, offering a rigorous yet intuitive framework that will be valuable to researchers across multiple fields. I commend the authors for their diligent revisions and believe the manuscript is now ready for publication.